# Characteristics of Primary Cutaneous Lymphoma in Italy: A Tertiary Care, Single-Center Study

Martina D'Onghia [1,*], Alessandra Cartocci [1], Laura Calabrese [1], Daniele Maio [1], Azzurra Sirchio [1], Maria Erasti [1], Linda Tognetti [1], Pietro Rubegni [1], Monica Bocchia [2], Emanuele Cencini [2], Alberto Fabbri [2] and Elisa Cinotti [1]

[1] Department of Medical, Surgical and Neurological Sciences, Dermatology Section, University of Siena, 53100 Siena, Italy; alessandra.cartocci@dbm.unisi.it (A.C.); laura.calabrese@unisi.it (L.C.); daniele.maio@student.unisi.it (D.M.); a.sirchio@student.unisi.it (A.S.); m.erasti@student.unisi.it (M.E.); linda.tognetti@dbm.unisi.it (L.T.); pietro.rubegni@unisi.it (P.R.); elisa.cinotti@unisi.it (E.C.)

[2] Department of Medical, Surgical and Neurological Sciences, Hematology Section, University of Siena, 53100 Siena, Italy; bocchia@unisi.it (M.B.); emanuele.cencini@ao-siena.toscana.it (E.C.); a.fabbri@ao-siena.toscana.it (A.F.)

* Correspondence: m.donghia@student.unisi.it

**Abstract:** Data on primary cutaneous lymphomas (PCLs) patients in the Italian population are limited, and, despite the existence of several treatment options, the management of those patients remains challenging. Our study aimed to investigate the clinical and therapeutic features of PCL patients in a referral center in Italy. We conducted a retrospective study on 100 consecutive PCL patients between January 2017 and December 2022. The mean (SD) age of our cohort was 70.33 (14.14) years. Cutaneous T-cell lymphomas (CTCLs) represented 65% of all cases; the majority were mycosis fungoides (42%), followed by cases of Sezary syndrome (10%) and primary cutaneous anaplastic large cell lymphoma (4%). Cutaneous B-cell lymphomas (CBCLs) accounted for 35 % of PCLs, with 15 cases of primary cutaneous follicle center lymphoma, 10 cases of primary cutaneous diffuse large B-cell lymphoma leg type, and 9 cases of marginal zone B-cell lymphoma. A higher frequency of pruritus ($p = 0.008$) and higher peripheral blood levels of beta-2 microglobulin ($p \leq 0.001$) and lactate dehydrogenase ($p = 0.025$) were found in CTCLs compared to those of CBCLs. Considering all therapeutic lines performed, treatments were extremely heterogeneous and skin-directed therapies represented the most frequently used approach. Our study confirms the distribution of PCL subtypes formerly reported in the literature and highlights the utility of real-life data in treatments to improve the current management of PCL patients.

**Keywords:** primary cutaneous lymphoma; skin; treatments

## 1. Introduction

Primary cutaneous lymphomas (PCLs) are a heterogeneous group of extra-nodal non-Hodgkin lymphomas arising from the malignant clonal transformation of T lymphocytes (cutaneous T-cell lymphoma, CTCL) or B lymphocytes (cutaneous B-cell lymphoma, CBCL). They are considered rare diseases that primarily proliferate in the skin and have no evidence of extracutaneous involvement at the time of diagnosis [1].

Evidence for PCL epidemiology is still limited, and recent studies have highlighted a growing incidence of both CTCLs and CBCLs [2], probably due to the better awareness of these entities and increased life expectancy [3]. Currently, studies from the United States have suggested an incidence of approximately 0.64–0.87/100,000 person-years, while researchers from Western Europe have described a lower incidence of 0.29–0.39/100,000 person-years [2]. Nevertheless, a recent French nationwide study reported an unprecedently high incidence of 0.96/100,000 person-years [3]. To the best of our knowledge, only few epidemiological reports have been conducted in Italy and showed a very low incidence (0.8/100,000 cases per year), with no sex differences [4].

Currently, the pathogenesis of PCL is complex and not fully understood [5]. Compared to systemic non-Hodgkin lymphomas, more than 75% of PCLs are CTCLs, while CBCLs account for approximately 30% of all subtypes [6]. As regards prognosis, while certain lymphomas are particularly aggressive with reduced survival, others have favorable survival profiles, characterized by multiple relapses and a chronic course [5].

PCLs generally present with a broad range of clinical manifestations and histopathological features that may mimic those of other benign skin conditions, posing a real diagnostic challenge to the dermatologist [7]. Moreover, molecular biology may not be particularly appropriate in the early stages of PCL, given the presence of few lymphoma cells in skin biopsies. Hence, PCL diagnosis is frequently delayed, and this fact could worsen patients' quality of life, leading to higher healthcare costs [8].

Although several guidelines and recommendations for PCL management exist, there is no general international consensus on PCL treatment [1,9–11]. Choosing the best therapeutic strategy is still challenging, and despite the wide range of potential therapies, PCL rarely results in a complete response [12,13].

In this context, a correct classification of PCL, along with the identification of independent prognostic factors, is crucial to drive clinical treatment decisions that can vary from skin-directed therapies (SDTs), such as phototherapy or topical corticosteroids (CS), to multiagent chemotherapy and hematopoietic stem cell transplantation. Hence, PCL management should require a multidisciplinary care team including a dermatologist, a hematologist–oncologist, and a pathologist.

Reliable literature on PCL in the Italian population is still lacking. Therefore, the primary aim of this study was to characterize the epidemiology and clinical features of PCL in a tertiary care center in Italy. The secondary aim was to describe treatment regimens and clinical responses in a real-life population.

## 2. Materials and Methods

We performed an observational retrospective analysis of 100 patients consecutively diagnosed with PCL at the Dermatologic Clinic of the University of Siena, Italy. In this study, all patients were included between January 2017 and December 2022. All diagnoses were confirmed both upon histopathological examination by the pathologists and by dermatologists and hematologists according to the World Health Organization (WHO)–European Organization for Research and Treatment of Cancer (EORTC) classification system [14]. New entities added to the revised WHO-EORTC classification in 2018 [1] were excluded from our analysis. The following characteristics were collected at the time of diagnosis: demographic data, diagnoses according to the 2005 WHO–EORTC classification [14], tumor–nodes–metastasis–blood (TNMB) staging at registration according to the International Society for Cutaneous Lymphomas (ISCL) and the Cutaneous Lymphoma–EORTC classification [15,16], the presence of pruritus, biochemical parameters, the treatments used, and the response to therapies. Clinical record descriptions and follow-up photo evidence were used to determine treatment responses, and no predetermined time interval was considered for the evaluation of responses to therapies. The mean patient follow-up period generally ranged from 3 to 12 months, according to the severity of the disease.

Descriptive statistics included the mean and standard deviation (SD) and median for continuous variables, whereas frequency and percentage were reported for categorical variables. To compare groups, the chi-squared test, Fisher's exact test, and Student's $t$ test were used. $p < 0.05$ was considered statistically significant. All data were assessed using R software version 4.1.0 (R Foundation for Statistical Computing, Vienna, Austria).

## 3. Results

### 3.1. Demographics and PCL Subtype Subsection

In total, 100 PCL patients were consecutively included between January 2017 and December 2022. The clinical characteristics and distribution of different PCL subtypes are summarized in Table 1. Briefly, 57 (57%) subjects were of the male sex. The mean ($\pm$ SD)

age at diagnosis was 70.33 (14.14) years. From our analysis, mature T-cell and natural killer (NK) cell lymphomas were the most common type of PCL (65%), followed by mature B-cell lymphomas (35%).

**Table 1.** General clinical and biochemical features, subtype, and stage of PCL.

| | Overall (*n* = 100) |
|---|---|
| Male, *n* (%) | 57 (57) |
| Age at diagnosis, mean (SD) | 70.33 (14.14) |
| PCL subtypes | |
| Cutaneous T-cell and NK-cell lymphomas | |
| Mycosis fungoides, *n* (%) | 42 (42) |
| Sezary syndrome, *n* (%) | 10 (10) |
| Primary cutaneous anaplastic large cell lymphoma, *n* (%) | 4 (4) |
| Primary cutaneous aggressive epidermotropic CD8+ cytotoxic T-cell lymphoma (provisional), *n* (%) | 3 (3) |
| Primary cutaneous CD4+ small/medium pleomorphic T-cell lymphoma, *n* (%) | 2 (2) |
| Subcutaneous panniculitis-like T-cell lymphoma, *n* (%) | 2 (2) |
| Extranodal NK/T-cell lymphoma, nasal type, *n* (%) | 1 (1) |
| Primary cutaneous peripheral T-cell lymphoma, unspecified *n* (%) | 1 (1) |
| Cutaneous B-cell lymphomas, *n* (%) | |
| Primary cutaneous follicle center lymphoma, *n* (%) | 15 (15) |
| Primary cutaneous diffuse large B-cell lymphoma, leg type, *n* (%) | 10 (10) |
| Primary cutaneous marginal zone lymphoma, *n* (%) | 9 (9) |
| Precursor hematological neoplasm, *n* (%) | |
| Blastic plasmacytoid dendritic cell neoplasm, *n* (%) | 1 (1) |
| TNMB stage at time of diagnosis for CTCL and CBCL * | |
| T1, *n* (%) | 33 (47.8) |
| T2, *n* (%) | 21 (30.4) |
| T3, *n* (%) | 4 (5.8) |
| T4, *n* (%) | 11 (15.9) |
| Clinical stage at time of diagnosis for MF and SS, *n* (%) ** | |
| IA, *n* (%) | 34 (73.9) |
| IB, *n* (%) | 5 (10.8) |
| IIA, *n* (%) | 1 (2.1) |
| IIB, *n* (%) | 0 (0) |
| IIIA, *n* (%) | 2 (4.3) |
| IIIB, *n* (%) | 0 (0) |
| IVA, *n* (%) | 4 (8.6) |
| IVB, *n* (%) | 0 (0) |
| Itch, *n* (%) | 22 (22) |
| LDH, mean (SD) | 211.8 (79.21) |
| LDH > ULN, *n* (%) | 46 (46) |
| Beta2-M, mean (SD) | 2.53 (1.50) |
| Beta2-M > ULN, *n* (%) | 77 (77) |

Legend: * TNMB staging was available for 69 patients. ** Clinical stage was available for 46 patients. Beta2-M, beta-2 microglobulin; CBCL, cutaneous B-cell lymphomas; CTCL, cutaneous T-cell lymphomas; LDH, lactate dehydrogenase; MF, mycosis fungoides; PCL, primary cutaneous lymphomas; SS, Sezary syndrome; TNMB, tumor–nodes–metastasis–blood; ULN, upper limit of normal.

The distribution of different PCL subtypes is shown in Figure 1. Among mature T-cell and NK-cell lymphomas, the most prevalent subtype was mycosis fungoides (MF) (*n* = 42), followed by Sezary syndrome (SS) (*n* = 10), primary cutaneous anaplastic large cell lymphoma (ALCL) (*n* = 4), primary cutaneous aggressive epidermotropic CD8+ cytotoxic T-cell lymphoma, provisional (*n* = 3) and primary cutaneous CD4+ small/medium pleomorphic T-cell lymphoma (SMPTCL) (*n* = 2), and subcutaneous panniculitis-like T-cell lymphoma (SPTL) (*n* = 2). Finally, one patient had extranodal NK/T-cell lymphoma of the nasal type, while another patient suffered from primary cutaneous peripheral T-cell

lymphoma (PTCL) of an unspecified type. Information on staging was not available for all patients. Most (76.9%) patients were diagnosed with early-stage disease (IA-IIA). Stage III or IV was observed in 7.7% and 15.9% of cases, respectively. The most common subtype of mature B-cell lymphomas was primary cutaneous follicle center lymphoma (PCFCL) (*n* = 15), followed by primary cutaneous diffuse large B-cell lymphoma (PDLBCL) of the leg type (*n* = 10), and primary cutaneous marginal zone B-cell lymphoma (PCMZL) (*n* = 9).

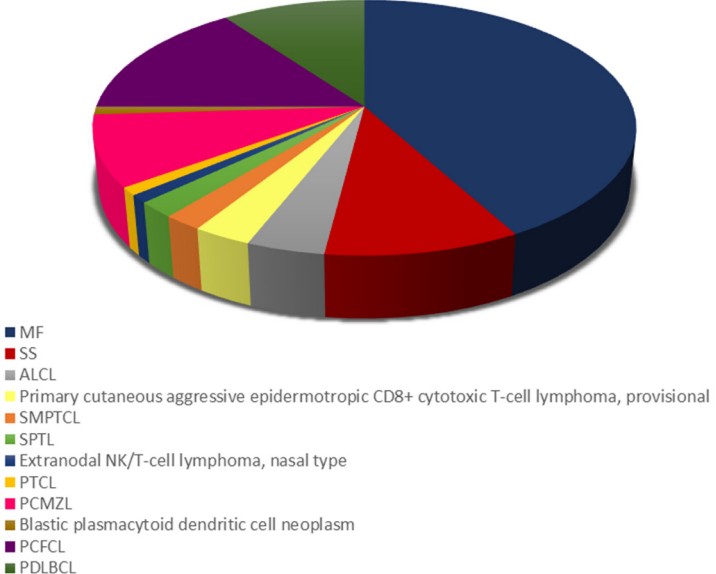

- MF
- SS
- ALCL
- Primary cutaneous aggressive epidermotropic CD8+ cytotoxic T-cell lymphoma, provisional
- SMPTCL
- SPTL
- Extranodal NK/T-cell lymphoma, nasal type
- PTCL
- PCMZL
- Blastic plasmacytoid dendritic cell neoplasm
- PCFCL
- PDLBCL

**Figure 1.** Distribution of cutaneous T-cell lymphoma (CTCL) and cutaneous B cell lymphoma (CBCL) subtypes. ALCL, primary cutaneous anaplastic large cell lymphoma; MF, mycosis fungoides; PCFCL, primary cutaneous follicle center lymphoma; PCMZL primary cutaneous marginal zone B-cell lymphoma; PDLBCL, primary cutaneous diffuse large B-cell lymphoma; PTCL, primary cutaneous peripheral T-cell lymphoma; SMPTCL, primary cutaneous CD4+ small/medium pleomorphic T-cell lymphoma; SPTL; subcutaneous panniculitis-like T-cell lymphoma; SS, Sezary syndrome.

We observed that CTCL tended to be diagnosed in older patients rather than CBCL patients (*p* = 0.002), while no sex differences were found between the two groups (see Table 2). Pruritus was reported by 22 (22 %) patients and was observed more frequently in the CTCL group (30.8%) than in the CBCL group (5.7%, *p* = 0.008). Considering the overall population at diagnosis, we observed that the mean (SD) beta-2 microglobulin (beta2-M) and lactate dehydrogenase (LDH) peripheral blood levels were 2.53 (1.50) ng/mL and 211.8 (79.21) mU/mL, respectively. Finally, we found higher peripheral blood levels of beta-2M and LDH in the CTCL group compared to those of CBCL patients, and also, we found a statistically significant difference between them (*p* ≤ 0.001 and *p* = 0.025, respectively).

**Table 2.** General population characteristics, pruritus, and biochemical parameters.

|  | **CBCL** | **CTCL** | ***p* Value** |
| --- | --- | --- | --- |
| Male, *n* (%) | 18 (51.4) | 39 (60.0) | 0.539 |
| Age at diagnosis, mean (SD) | 64.57 (13.56) | 73.43 (13.55) | 0.002 |
| Itch, *n* (%) | 2 (5.7) | 20 (30.8) | 0.008 |
| LDH, median (IQR) | 181.00 (156.00, 200.00) | 200.00 (173.50, 245.00) | 0.025 |
| Beta2-M, median (IQR) | 1.75 (1.50, 2.18) | 2.40 (2.00, 3.22) | <0.001 |

Legend: Beta2-M, beta-2 microglobulin; CBCL, cutaneous B-cell lymphoma; CTCL, cutaneous T-cell lymphoma; LDH, lactate dehydrogenase; IQR, Interquartile range; SD, standard deviation.

*3.2. Treatment Modalities*

An overview of all the treatments performed is presented in Figure 2. Our analysis referred to all treatment lines performed, and treatment responses were related to the last follow-up visit. During the study period, 18 different treatment regimens were observed, including skin-directed therapies (SDTs), systemic treatments (STs), surgery excision, and the "watch and wait"(WW) strategy. In addition, SDTs were divided into topical corticosteroids (CSs), phototherapy, and radiation therapy, which included radiotherapy (RT) and total skin electron beam therapy (TSEB). Systemic treatments included retinoids (bexarotene and acitretin), systemic CSs, extracorporeal photopheresis (ECP), monoclonal antibodies (rituximab, brentuximab-vedotion, and mogamulizumab), and chemotherapy (CT) (see Table 3).

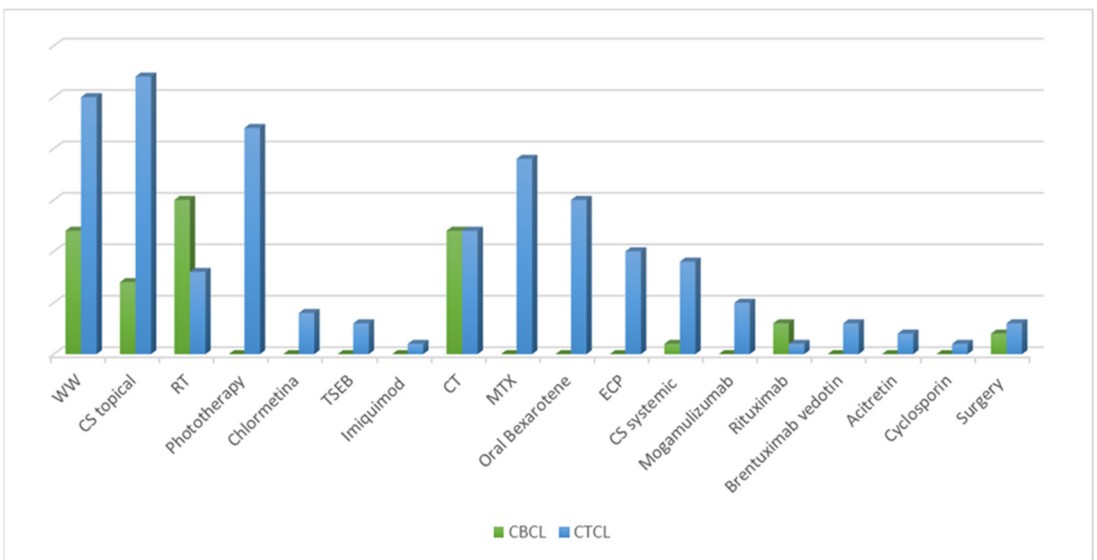

**Figure 2.** Analysis of all the treatment lines performed during the follow-up, divided according to cutaneous T-cell lymphoma (CTCL) and cutaneous B-cell lymphoma (CBCL). CS, corticosteroids; CT, chemotherapy; ECP, extracorporeal photopheresis; MTX, methotrexate; RT, radiotherapy; TSEB, total skin electron beam therapy; WW, watch and wait.

Overall, SDTs represented the most frequent therapeutic approach. Topical CSs were the most common treatments (*n* = 35), followed by RT (*n* = 23) and phototherapy (*n* = 22). The highest percentage of patients achieving a complete response (RC, 61.1%) was observed among those undergoing RT. Regarding ST, CT was primarily used (*n* = 24), followed by methotrexate (MTX) (*n* = 19) and oral bexarotene (*n* = 15). Ten patients underwent ECP. Only one patient with a diagnosis of panniculitis-like lymphomas was treated with cyclosporin. In 37 cases, we adopted the WW policy, while 5 patients underwent surgery, achieving a partial response in 75% of cases. Overall, we found that all treatment modalities evaluated during follow-up resulted in a high overall response rate.

Considering CTCL, we found that the most frequently used therapy was topical CS (*n* = 27), followed by phototherapy (*n* = 22), MTX (*n* = 19) and oral bexarotene (*n* = 15). The recently introduced topical chlormethine gel was used in four MF patients. New systemic drugs such as mogamulizumab and brentuximab-vedotin were administered to five and three patients, respectively. Interestingly, partial response (RP) was recorded for all patients that received mogamulizumab, including one patient who had previously received a heart transplantation.

**Table 3.** General treatment characteristics and analysis of achievement of complete or partial response.

| | Overall *n* | CBCL | CBCL, RP *n* (%) | CBCL, RC *n* (%) | CTCL | CTCL, RP *n* (%) | CTCL, RC *n* (%) | Overall Follow-Up Median (IQR) |
|---|---|---|---|---|---|---|---|---|
| W&W | 37 | 12 | 3 (75) | 1 (25) | 25 | 2 (11.8) | 10 (58.8) | 7.23 (3.73, 12.73) |
| **Skin Direct therapies** | | | | | | | | |
| Topical CS | 35 | 7 | 3 (71.4) | 2 (28.6) | 27 | 9 (60) | 6 (40) | 6.07 (3.12, 11.38) |
| RT | 23 | 15 | 4 | 9 | 8 | 3 (60) | 2 (40) | 0.97 (0.62, 7.35) |
| Phototherapy | 22 | 0 | Na | Na | 22 | 10 (66.7) | 5 (33.3) | 5.02 (3.10, 8.96) |
| Chlormethine | 4 | 0 | Na | Na | 4 | 2 (100) | 0 (0) | 5.78 (3.68, 7.95) |
| TSEB | 3 | 0 | Na | Na | 3 | 3 (100) | 0 (0) | 0.97 (0.82, 2.13) |
| Imiquimod | 1 | 0 | Na | Na | 1 | 0 (0) | 0 (0) | 0.93 (0.93, 0.93) |
| **Systemic therapies** | | | | | | | | |
| CT | 24 | 12 | 4 (33.3) | 8 (66.7) | 12 | 10 (90.9) | 1 (9.1) | 4.13 (3.22, 6.07) |
| MTX | 19 | 0 | Na | Na | 19 | 12 (85.7) | 2 (14.3) | 6.33 (2.55, 13.65) |
| Oral bexarotene | 15 | 0 | Na | Na | 15 | 12 (85.7) | 2 (14.3) | 10.27 (5.52, 23.53) |
| ECP | 10 | 0 | Na | Na | 10 | 7 (77.8) | 2 (22.2) | 18.42 (5.28, 35.62) |
| Systemic CS | 10 | 1 | 1 (100) | Na | 9 | 5 (71.4) | 3 (42.9) | 5.15 (2.68, 13.96) |
| Mogamulizumab | 5 | 0 | Na | Na | 5 | 5 (100) | 0 (0) | 0.00 (0.00, 0.00) |
| Rituximab | 4 | 3 | 2 (66.7) | 1 (33.3) | 1 | 0 (0) | 1 (100) | 1.65 (0.00, 5.51) |
| Brentuximab-vedotin | 3 | 0 | Na | Na | 3 | 2 (66.7) | 1 (33.3) | 0.97 (3.90, 4.23,) |
| Acitretin | 2 | 0 | Na | Na | 2 | 2 (100) | 0 (0) | 13.58 (10.34, 16.82) |
| Cyclosporine | 1 | 0 | Na | Na | 1 | 1 (100) | 0 (0) | 12.13 (12.13, 12.13) |
| Surgery | 5 | 2 | 1 (50) | 1 (50) | 3 | 2 (100) | 0 (0) | - |

Legend: CBCL, cutaneous B-cell lymphoma; CS, corticosteroids; CT, chemotherapy; CTCL, cutaneous T-cell lymphoma; ECP, extracorporeal photopheresis; MTX, methotrexate; Na, Not Available; RT, radiotherapy; TSEB, total skin electron beam therapy; WW, watch and wait. All percentages were calculated for patients with available treatment responses.

In our CBCL cohort, treatments mainly included RT (*n* = 12) and CT (*n* = 12). In seven CBCLs, we used topical CSs and precisely, we conducted the intralesional injection of tri-amcinolone acetonide combined with lidocaine with epinephrine in a ratio of 1:100,000 and with saline in a ratio of 1:1:2. Rituximab was used in four cases. Notably, in CBCL patients, the overall RC ranged from 25% to 66.7%.

Furthermore, combinations of therapies were observed. The most common combined treatment was topical CSs in addition to other systemic therapies, although it was difficult to estimate the number of patients with these combinations over time. In our analysis, we focused on CTCL patients receiving MTX or bexarotene. Overall, six patients received MTX combined with a topical CS (*n* = 3), a systemic CS (*n* = 1), phototherapy (*n* = 1), and ECP (*n* = 1). We commonly used low-dose MTX at 15 mg/week. Regarding bexarotene, 12 patients received this drug combined with ECP (*n* = 8), systemic CS (*n* = 2), TSEB (*n* = 1) or CT (*n* = 1).

## 4. Discussion

Cutaneous lymphomas are a heterogeneous group of rare malignancies with various presentations, in which the skin is the primary organ involved. Reports on PCL in Italy are still limited; hence, the goal of our study was to present some original findings on PCL epidemiological characteristics, clinical outcomes, and treatments in an Italian cohort. Our results showed a mean (SD) age at diagnosis of 70.33 (14.14) years, confirming that PCL predominantly affects adults [2]. Moreover, a statistically significant difference was found between CTCL and CBCL in terms of age (*p* = 0.002), underlining that CTCL patients are usually older [17]. In accordance with the current literature, we also observed an overall global male predominance (57%) [2].

In our study, we found that CTCL accounted for 65% of patients, with MF being the most prevalent subtype (*n* = 42), followed by SS (*n* = 10). This finding is not surprising because an extensive body of literature has demonstrated that MF and SS are the most common entities, representing more than half of all PCLs, followed by CD30-positive lymphoproliferative disorders [5]. Significant differences have been reported in PCL relative frequencies between continents and even within Europe [2]. However, our analysis resembles the CTCL distribution formerly reported in other Western countries, such as Germany [18] and the United States [19], but also in Korea [20]. In addition, our data consistently confirm the findings of previous national reports [4,21,22].

When focusing on CBCL, we found that the most common subtype of mature B-cell lymphomas was PCFCL (*n* = 15), followed by PDLBCL (*n* = 10) and PCMZL (*n* = 9). The high prevalence of PCFCL agreed with current evidence, revealing that it represents approximately 55% of all CBCL [23]. Conversely, we found a frequency of PDLBCL that was slightly higher than that of PCMZL, which differs from what is generally reported [4,18,21]. The reasons underlying those differences are unclear, and both genetic and environmental factors may be involved. However, another intuitive explanation could be the small sample size in the studied cohort, since PCLs are extremely rare entities.

In our study, the majority of patients (76.9%) were diagnosed at an early stage (IA-II), and PCL other than MF or SS (47.8%) mostly presented at T1 according to the TNM classification [15]. Those data are consistent with those in earlier publications [3,22].

Interestingly, 22% of our patients had pruritus at diagnosis, and a significant difference between CTCL and CBCL groups was reported (*p* = 0.008). Pruritus is a symptom that critically affects quality of life, and it is often associated with hematological malignancies [24]. CTCL, especially in advanced stages, can cause intractable pruritus in approximately 62% to 88% of cases [25]. Furthermore, pruritus may be associated with CBCL in approximately half of patients [26]. The pathophysiology behind malignancy-associated itch remains elusive. Investigations into the pruritogenic components of CTCL have implicated a complex interplay between different mediators, including inteleukin-31, nerve growth factors, and substance *p* as putative mediators. It should be notated that the presence of pruritus may drive physicians' treatment decisions, particularly in the early stages, where it could

be the only symptom. Moreover, it has been demonstrated that worsening pruritus is associated with disease progression [25], and an Italian study has recently outlined that pruritus may represent an important clinical parameter to be considered a predictive factor of clinical response [21]. The identification of reliable predictive factors may contribute to the choice of the best therapeutic regimens for each patient. However, although several PCL prognosticators have been studied, the effectiveness of their role is still debated.

Peripheral blood levels of beta-2M and LDH are increased in many tumors, including non-Hodgkin lymphomas [26]. Our results showed a higher mean level of beta-2M (2.53, SD1.50), while mean LDH (211.8, SD 79.21) levels were in the normal range. However, our results need to be analyzed carefully given the heterogeneity of the entities considered. Several studies have demonstrated that LDH may be associated with later tumor stages and worse prognosis, particularly in MF and SS [27], whereas only hints of evidence of the role of beta-2M levels are present for cutaneous lymphomas [28]. Considering peripheral beta-2M and LDH levels, we observed a statistically significant difference between CTCL and CBCL ($p = 0.025$ and $p < 0.001$, respectively). Even if this association had not been extensively studied, we could speculate that patients with CTCL could have a higher tumor burden.

In this study, we attempted to summarize treatment approaches in a PCL cohort from a real-world perspective, referring to all treatment lines performed. Over the years, notable changes were observed in treatment approaches. The first finding from our results was that treatment strategies were characterized by striking heterogeneity with up to 18 different drugs, alone or combined. Our results mirror the worldwide situation, and are mainly explained by the rarity of those malignancies, which impaired the design of clinical trials and thus the development of homogeneous treatment guidelines [29]. This means that treatment algorithms may vary according to the physician's preference or skills, the institute's equipment, or the availability of treatment modalities. Although we did not report the association between specific treatment approaches and disease stages, the latter clearly remains one important parameter for choosing the most suitable therapies [30,31]. Secondly, we found that all treatment modalities evaluated during follow-up resulted in a high overall response rate (see Table 3).

Overall, the "watch and wait" policy was chosen in 37 cases. Broadly speaking, this strategy can be considered for stage IA mycosis fungoides, primary cutaneous CD4+ small/medium pleomorphic T-cell lymphomas, and subcutaneous panniculitis-like T-cell lymphomas with no symptoms. However, we also adopted this strategy if the lymphoma was asymptomatic, localized, and had an indolent course. Considering CTCL, we observed that SDTs represented the most frequent therapeutic approach, particularly topical corticosteroids ($n = 27$), followed by phototherapy ($n = 22$). Skin-directed therapies are the mainstay of treatment in the early stages of PCL, including MF, which was the most common subtype in our population. Our patients were predominantly treated with topical corticosteroids, which emphasizes their importance both as a single therapy and as an adjuvant in addition to ST [32]. Notably, the use of phototherapy represented a viable treatment option; thus, we found that phototherapy was the second SDT prescribed for CTCL ($n = 22$). This treatment is frequently used in managing patients with MF and lymphomatoid papulosis, with high complete remission rates but variable response durations [32]. Recently, STs are becoming more commonly used for advanced stages of CTCL, previous treatment failure with SDT, or refractory cases. From our analysis, low-dose MTX ($n = 19$) and oral bexarotene ($n = 15$) were the most commonly used STs, either alone or in combination. One patient with a diagnosis of SS received MTX combined with systemic steroids, whereas in eight cases, SS was treated with bexarotene and ECP. Other treatments included new drugs, such as brentuximab ($n = 3$) or mogamulizumab ($n = 5$), used as treatment options for relapsed or refractory cases, since they do not represent a first-line therapy. In our CBCL cases, treatment modalities were mainly represented by RT ($n = 12$) and CT ($n = 12$), followed by topical corticosteroids ($n = 7$) and rituximab ($n = 4$). CBCLs are often localized in small areas of the skin and mainly respond to SDTs, including surgery, RT and topical CSs [33]. The

medical literature reported a high efficacy of RT, allowing complete or partial remission to be achieved in more than 95% of patients [34], data that were confirmed by our preliminary data. In line with current evidence, we decided to treat the generalized form of CBCL systematically, mainly through CT [35], reaching a complete response in 40 % of cases.

The main strength of this work lies in the accuracy of the diagnosis and treatments performed, in a relatively large sample of Italian PCL patients. Recently, Pileri et al. [22] presented some data retrieved from the cutaneous lymphoma registry of the Italian Marche region, emphasizing the need for a national PCL registry, since to date no detailed epidemiological data exist in Italy. Thus, a national registry could allow for more comprehensive data collection across the country, providing information on the incidence and epidemiology of those rare malignancies. Our analysis could contribute to the clarification of some epidemiological and clinical aspects of PCL in Italy and may help in the future in the development of predictive therapeutic algorithms based on real-life data. However, several limitations need to be recognized, including the retrospective nature of our study, which may be responsible for some missing data on clinical information or follow-up data, affecting our results. Moreover, given the heterogeneity of therapeutic options we used over time, alone or in combination, we could not report the precise relation between PCL subtypes and treatments performed. Finally, a direct comparison between Italian studies was complicated by the lack of a consistent body of literature on our geographic area, and by the differences in PCL classification and treatment strategies used in previous national studies.

## 5. Conclusions

Primary cutaneous lymphomas encompass a wide variety of lymphomas. Our results highlight the unmet need for large population-based cohort studies, which may be fundamental to improving the current knowledge about those malignancies and could lead to early diagnosis and better tailored management via choosing the best therapeutic approach for each patient.

**Author Contributions:** Conceptualization, M.D. and E.C. (Elisa Cinotti); methodology, M.D.; software, A.C.; validation and formal analysis, A.C.; investigation, A.S., M.E. and A.F.; resources, D.M.; data curation, M.D.; writing—original draft preparation, M.D. and E.C. (Elisa Cinotti); writing—review and editing, P.R., E.C. (Emanuele Cencini), L.C. and A.F.; visualization, L.T., L.C. and M.B.; supervision, P.R. All authors have read and agreed to the published version of the manuscript.

**Funding:** This research received no external funding.

**Institutional Review Board Statement:** Ethical review and approval were waived for this study because it is an observational and retrospective study that did not change our clinical practice.

**Informed Consent Statement:** Informed consent was obtained from all subjects involved in the study.

**Data Availability Statement:** Data are available upon request to the corresponding author.

**Conflicts of Interest:** The authors declare no conflict of interest.

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
