# Peer review of "Characteristics of Primary Cutaneous Lymphoma in Italy: A Tertiary Care, Single-Center Study"

_curroncol, doi:10.3390/curroncol30110712_

Round 1

Reviewer 1 Report

Comments and Suggestions for Authors
  1. The introduction is well-structured and clear. It begins with a brief explanation of PCL, its rarity, and its classification into T-cell lymphomas (CTCL) and B-cell lymphomas (CBCL). The article provides valuable insights into the study of PCL in Italy, but some improvements in clarity and presentation could enhance its quality.

I suggest the areas that can be improved: Interpret the results in the context of existing research, explaining how they contribute to our understanding of PCL. Discuss any study limitations that might affect the interpretation of the results.

Author Response

We thank the reviewer for the comment. According to the reviewer's comment we modified the discussion of our Manuscript, elaborating strength and limitation of the study more accurately.

Reviewer 2 Report

Comments and Suggestions for Authors

D’Onghia and Coworkers report on the experience of an Italian Centre in the field of primary cutaneous lymphoma (PCL). On this respect, the contribution is of interest since only a few attempts have been made in Italy to assess the prevalence of these neoplasms as well as of the different lymphoma entities. It might be reminded the existence of a registry of PCL in the Marche region. The manuscript suffers from an inherent contradiction. In fact, in the introduction the Authors claim that “PCL management should require a multidisciplinary care team including a dermatologist, a hematologist-oncologist, and a pathologist”. However, in the Materials and Methods, they state that “all diagnoses were confirmed by both dermatologists and hematologists”. In fact, no pathologist is involved in the manuscript, although most if not all diagnoses require microscopic evaluation, immunohistochemical tests, and molecular studies in selected cases. Thus, the entire series should be diagnostically validated by a pathologist. This is mandatory to further evaluate the manuscript.

Comments on the Quality of English Language

See RP and RC instead of PR and CR.

Author Response

We thank the reviewer for the comment. Firstly, we have mentioned in our paper the PCL registry of the Marche region by Pileri et al., as you suggested. Secondly, In the Introduction we claimed that the pathologist is fundamental in the multidisciplinary team, since, as you pointed out, PCL diagnoses require most of the time microscopic evaluation, immunohistochemical tests, or molecular studies. Indeed, all our patients had a diagnosis on histopathological examination, thus confirming the determinant role of the pathologist.  However, in the Materials and Methods section we only mentioned “dermatologists and haematologists” since they are responsible for providing direct care to PCL patients from a clinical point of view and during the follow up period.  However, to better clarify this point, we added this statement in the Materials and Methods section.

Reviewer 3 Report

Comments and Suggestions for Authors

The authors retrospectively summarized the epidemiology and clinical features of 100 cutaneous lymphoma patients in a tertiary care center in Italy. Cutaneous lymphomas are rare diseases, and such an epidemiologic study is important. I have some concerns before the publication.

1. Please check Table 1 carefully. The lines of the number of patients seemed to be different from those of disease names. 

2. Was lymphomatoid papulosis excluded from this study? Lymphomatoid papulosis is usually included in the prevalence study of cutaneous lymphomas.

3. Figure 1. Which color is PCDLBCL? The explanation may be missed. “ACLC” may be “ALCL”.

4. Table 3. Cutaneous lymphomas include various subtypes that have the different characteristics from each other. The overall RP and RC in the whole cohort of each therapy are meaningless and should be removed.

5. I think wait and watch policy is selected in so many cutaneous T-cell lymphoma cases in the authors’ cohort. The policy can be considered for stage IA mycosis fungoides, primary cutaneous CD4+ small/medium pleomorphic T-cell lymphoma, and subcutaneous panniculitis-like T-cell lymphoma with no symptoms. Is there any reasonable explanation?

Author Response

We thank the reviewer for the comments.

  1. We have correctly realigned the lines of the number of patients with the disease names in Table 1.
  2. Although lymphomatoid papulosis was not excluded from the prevalence study of cutaneous lymphoma, we did not have any patients with this diagnosis (as you can see from Table 1). This is probably due to the fact that we are a tertiary care centre, and we receive patients that are already seen by other dermatologists/ haematologists. It is likely that LF patients were not referred to our centre due to the spontaneous resolution of this PCL subtype.
  3. We have corrected figure 1 based on what you suggested (“PCDLBCL” was added and “ACLC” replaced with “ALCL”)
  4. We are aware that cutaneous lymphoma include various subtypes that have different characteristics from each other, which makes our data about “overall partial or complete response” only partially relevant. However, we thought that showing this data could emphasize how, despite the heterogeneity of treatments and no consensus about the management of PCL patients, we obtained a high rate of clinical response (partial or complete). However, to be more accurate, we have decided to divide complete and partial response according to cutaneous B cell lymphoma and cutaneous T cell lymphoma.
  5. We presented an analysis of all the treatment lines performed during the follow up period. A high rate of wait and watch policy could be attributed not only to cases such as stage IA mycosis fungoides, primary cutaneous CD4+ small/medium pleomorphic T-cell lymphoma (that represents a significant proportion of our cohort), but also to the fact that we chose “watch and wait” policy in all those PCL patients that presented an asymptomatic localized and indolent (non-progressive) course. Moreover, in this group we included the patients that rarely use emollients or topical steroids and who are not receiving topic or systemic treatments not on daily basis.

Round 2

Reviewer 1 Report

Comments and Suggestions for Authors

No comments, accept in the present form. 

Author Response

We thank the reviewer. 

Reviewer 2 Report

Comments and Suggestions for Authors

Although the Authors state that all cases were histologically reviewed, it seems a bit unorthodox that there are no pathologists among the Co-workers. This also in the light of the fact that the Authors claim that there is the need for integration among dermatologists, hematologists and pathologists.

Comments on the Quality of English Language

English revision can be made by the Editor.

Author Response

Although the pathologists histologically reviewed our cases, they did not fulfill the criteria for the authorship (i.e. conceptualization; methodology; software; etc.), for that reason they are not among co-workers. 

Reviewer 3 Report

Comments and Suggestions for Authors

The authors revised the manuscript well and the responses are convincing. However, I still have one minor concern. Based on the presented PDF file, the lines of the number of patients seemed to be till different from those of disease names in Table 1. The number of cutaneous T-cell and NK-cell lymphoma patients seemed to be 42 and that of mycosis fungoides seemed to be 10. Are they correct?

Author Response

We thank the reviewer for the comment. To further clarify what you requested, please see the attachment. 
